# Using Visualization to Build Transparency in a Healthcare Blockchain Application

**Jesús Peral [1,\*], Eduardo Gallego [1,2], David Gil [2], Mohan Tanniru [3] and Prashant Khambekar [4]**

[1]   Department of Software and Computing Systems, University of Alicante,
      03690 Alicante, Spain; ejgl2@alu.ua.es
[2]   Department of Computer Technology and Computation, University of Alicante,
      03690 Alicante, Spain; dgil@dtic.ua.es
[3]   College of Public Health, University of Arizona, Phoenix, AZ 85006, USA; tanniru@oakland.edu
[4]   Harbinger Systems, Philadelphia, PA 19103, USA; Prashant.Khambekar@harbingergroup.com
\*   Correspondence: jperal@dlsi.ua.es; Tel.: +34-96-590-3772

**Abstract:** With patients demanding services to control their own health conditions, hospitals are looking to build agility in delivering care by extending their reach into patient and partner ecosystems and sharing relevant patient data to support care continuity. However, sharing patient data with several external stakeholders outside a hospital network calls for the development of a digital platform that is trusted by both hospitals and stakeholders, given that there is often no single entity supporting such coordination. In this paper, we propose a methodology that uses a blockchain architecture to address the technical challenge of linking disparate systems used by multiple stakeholders and the social challenge of engendering trust by using visualization to bring about transparency in the way in which data are shared. We illustrate this methodology using a pilot implementation. The paper concludes with a discussion and directions for future research and makes some concluding comments.

**Keywords:** blockchain; IoT; secure transaction; health; file sharing; visualization

---

## 1. Introduction

In today's digital age, advanced technologies are continually altering customer expectations of services delivered and requiring that organizations build "agility" within their internal operations by using an agile organizational model of structure and governance [1]. The agile model supports the exploration of innovative service value propositions and the use of a mix of internal and external resources to evaluate these innovations to fulfill customer value [2,3]. Such a model is also used to support evaluation, adaptation, and learning to improve organizational capacity to sustain value as customer expectations change [4,5]. One can argue that "agile" organizations are indeed sustainable organizations, as they continue to meet the current needs of customers by using external resources and conserve their own resources to address future customer needs. In this paper, we focus on hospitals that are responsible for supporting continuity of care for patients outside the hospital.

Hospitals are extending patient care using several care facilities (e.g., urgent care facilities, ambulatory care facilities, etc.) [6,7] and are helping patients self-manage their care using multiple technologies [8–11]. This calls for hospitals to build agility to leverage the resources of external partners and motivate patients to self-manage their health in a tightly regulated and resource constrained environment. This means that patient data are generated by multiple stakeholder systems (partners and patients) that use several advanced technologies, such as internet of things (IoT), mobile apps, digital exchanges, and social media, and such data have to be understood, collected, integrated, and shared by all involved in the support of patient care. Unless there is a public health crisis

(e.g., COVID-19) that calls for public health agencies to coordinate significant disruptions to economic, health, and social conditions [12], opioid addiction that calls for tracking drug distribution [13], or chronic care management of high risk patients to reduce hospital readmissions [14–16], there is little incentive for hospitals to coordinate patient data sharing outside their hospital networks. This calls for a distributed digital platform that is either coordinated by a trusted third party or an architecture that ensures trust for everyone to contribute and use the data shared.

Blockchain technology has been suggested in prior research as a platform when there is no trusted coordinator to support data sharing. It supports peer-to-peer connectivity among various stakeholders using agreed upon protocols about who can participate in such data sharing. Using characteristics such as immutability and auditability, it is considered a viable and trusted platform to share data when there is no central entity coordinating such sharing activities. In healthcare, establishing trust is both a technical challenge (i.e., ensuring the integrity of data shared by multiple stakeholder systems and making it available for impact on care) and a social challenge (i.e., ensuring transparency to engender confidence that the mechanism used to share data addresses confidentiality). For example, a system that monitors patients' vital signs and uses an algorithm to generate a metric used to track patient conditions has to be trusted for its integrity. Patients' questions sent to peers and clinicians may be anonymized to share with peers for comment to ensure confidentiality and identified and made available to patients quickly for treatment adaptation. This paper proposes a methodology that uses blockchain technology as a digital platform with a visualization feature added to address both the technical and social challenges.

This paper is organized as follows. Section 2 provides prior research on the use of blockchain in multiple domains as well as in healthcare. Section 3 discusses the methodology that creates visibility for both the creation of the data and its movement within the network to address the challenges. A case study to illustrate a pilot implementation is explained in Section 4. Section 5 discusses, in detail, an implementation methodology, and Section 6 includes discussion, future research directions, and limitations. Section 7 provides some concluding comments.

## 2. Background

Blockchain applications can be categorized by domain-financial or non-financial [17], since cryptocurrencies represent many but not all of the applications using blockchain technology. These applications can also be classified by the version of technology used (i.e., 1.0, 2.0 and 3.0) [18,19]. Along application domains, they can classified by application type (e.g., financial, healthcare, business and industrial, education, etc.), business issue focus (e.g., governance, privacy and security, etc.), or technical issue focus (e.g., integrity verification, IoT, data management, etc.) [20]. Application of blockchain in healthcare has been more recent [21,22], and, as discussed earlier, trust in sharing sensitive healthcare information among several actors outside a hospital system has been a challenge [23]. However, the mechanisms embedded in the distributed ledger technology associated with blockchain technology may be able to address this challenge [21,22,24–26]. In other words, if healthcare organizations are to become agile in meeting patient needs outside a hospital, the digital platform has to address some of the technical challenges, such as ensuring security, interoperability, data sharing, and mobility, if it is to engender trust [23]. Let us take each of these in more detail.

(1) Security

Existing methods used to protect and secure patient medical records have not been effective [27,28]. While access controls and authentication of records are widely used in ensure integrity, confidentiality, and accessible of medication information [26,29,30], their implementation becomes a challenge once systems are extended outside a hospital [31,32]. The encryption of data among Electronic Medical Record (EMR) and stakeholder systems is useful, but this leads to problems when there are many different encryption standards [33,34]. With no single technology platform addressing the security challenges [35], a distributed platform that allows local control of the data at each node but ensures

security as it moves across a distributed platform may be a solution. Blockchain technology, which has a uniform method to encrypt the data transferred, public–private keys for the authentication of users who transfer the data, and validation of those who decrypt the data for use, can be effective in addressing security when data are shared by several stakeholders [20,26,36,37].

(2) Interoperability

Sharing data among multiple stakeholder systems, such as apps or intelligent agents, or multiple people, such as messages sent using mobile phones, requires having a uniform method to collect disparate sources of data and a centralized database for all to share. With no single entity coordinating such a shared database, a blockchain architecture can allow each partner to upload data for sharing and use using certain agreed upon protocols about who can contribute and access data, with embedded security, controlled redundancy, and auditability [23].

(3) Data Sharing

Data sharing in healthcare is critical, as patient care is remotely managed at various locations (at home, at partner sites, or at hospitals) and must be shared with others to support continuity of care. Moreover, the data gathered at each site may be in a different form [24,33,38]. Blockchain technology allows for each partner connected to the network to share data either directly or indirectly using a secure link. In some cases, data are stored elsewhere (e.g., when the data are large, as with image scans, or in narrative form, as with doctors' notes), and associated links can be used for data access. In summary, blockchain technology allows the sharing of multiple data types without forcing a single data normalization method.

(4) Mobility (IoMT)

As patients become mobile and must access their data when and where they need it, its portability is critical. With more devices such as smart phones and sensors (IoT) connected to the Internet, data are collected from these devices [39–41] have to be effectively integrated. This concept is often referred to as digital mobility or Internet of Medical Things (IoMT) [42–44]. With a blockchain's ability to connect with any partner (human or machine) with permission to share data with others, such mobility is feasible.

*2.1. Blockchain in Healthcare*

Blockchain technology has begun to see applications that extend care to patients outside a hospital. Traditionally, EMRs are used to manage patient data within a hospital system, and their use has grown significantly [10,45,46]. However, as hospitals try to extend care to patients outside the hospital, and with partners and patients using a myriad of systems, the challenge is one of interoperability. Blockchains can provide a gateway for data sharing among these systems by addressing the four key areas of importance discussed above: security, interoperability, data sharing, and mobility. For example, OmniPHR (Omnipresent Personal Health Record) has been proposed as a distributed model to integrate personal health records for patients and hospitals to access and use [38], and MedRec (decentralized record management system to handle EMRs) is being developed as a component of a hospital EMR system [24]. A framework for EMR data sharing for cancer patients is proposed by Dubovitskaya et al. [47], and a decentralized platform that provides a secure, fast, and transparent exchange of a single version of a patient's data are provided by Medicalchain [48].

Other applications include HealthChain, which leverages blockchain technology to support the sharing of patients' medical data [49,50]. MediBchain is another patient centric healthcare data management system that enables patient data sharing using cryptographic techniques [51]. Borioli and Couturier [52] discuss the potential of blockchain to conduct clinical trials using smart contracts, and Mamoshina et al. [53] propose a roadmap for decentralizing the personal health data ecosystem for drug discovery, biomarker development, and preventative healthcare. The use of microscopy sensors

that take an image of fingernails for identity authentication was proposed by Lee at al. [54] to protect data privacy, and an Ethereum protocol that remotely monitors and manages patients using data from sensors and smart devices and smart contracts was presented by Griggs et al. [27]. MeDShare (a system that addresses the issue of medical data sharing) is a blockchain-based system that is used to provide data provenance, auditing, and control of shared medical data in cloud repositories and to monitor malicious use of these data [28].

The goal of all these applications is to support operational continuity as care is extended outside a hospital so that patient data can be accessed by doctors, hospitals, laboratories, pharmacists, insurers, etc., and strategic support (e.g., analysis for treatment adherence to change diagnoses or treatment plans, at an individual level over time or at an aggregate level for discovering patterns, possibly using big data analytics). To address these two types of support, one may consider two different blockchain architectures: one blockchain with parallel computing capabilities and big data analytics for strategic support, and another blockchain to support operational continuity that includes data integration, secure identity management, and a trust supporting data sharing component [55]. Each of these blockchains still leverages the blockchain properties of authentication, confidentiality, accountability, and data sharing among those using the networks. In other words, operational continuity leads to data collection (or surveillance of patient–partner activities), and strategy support is used to leverage these data for analysis and to refine care processes.

## 2.2. Increasing Trust through Visibility

While the discussion thus far demonstrates the role of blockchain technology in addressing a number of technical challenges to ensure trust in the way data are collected from disparate systems and shared to ensure integrity and confidentiality, there is still the issue of the social challenge: Will those who have to adopt the system trust the system enough to contribute to it? Transparency through visualization to enhance trust has been discussed in the literature. For example, transparency of the supply chain is viewed as critical to engender trust among the participating stakeholders [56], and visualization is often used to communicate information to groups with varying technical backgrounds, especially when there are opportunities for misrepresentation of the data [57]. In some cases, interactive graphics are used to make static reports dynamic, so that individuals can understand the data by seeing such data at various levels of granularity [58]. Dashboards with drill-down capabilities have been used by many organizations to improve both transparency and accountability, especially when clinical decisions and administrative decisions lead to conflicts [59].

Visualizations has also been used to debug software and help with understanding the reasoning processes of forward-chaining rule-based expert systems [60], as well as when individuals are engaged in global software development to ensure that workflows that are generating data to influence a project can be monitored [61]. Today, when data are manipulated by multiple entities including robots, designing human-like and visualization-based transparency is critical to map the processes used to manipulate data so it can match an individual's mental models [62] and reduce the cognitive burden by helping with external anchoring, information foraging, and cognitive offloading [63]. The methodology discussed here uses "visualization" to improve the trustworthiness of those sharing the data using the blockchain architecture, thus addressing both technical and social challenges.

## 3. The Proposed Methodology: A Blockchain-Based Solution

In this section, we present our general methodology where a blockchain architecture is used to visually show how data are shared by users as it moves among various nodes in the network. The architecture uses two web applications: one to create the data for the blockchain and the other to visualize the network to improve transparency and build trust. The application supports the sharing of data files (PDF, text, images, etc.) between different nodes, so that a user will have the ability to visually see the files as they are sent and received, ensuring the existence, order, and immutability of these files. Specifically, we will illustrate the process used when permission is granted for some data by the patient

and the subsequent movement of these data along the network to support transparency. To achieve the stated objectives, the methodology uses two features: blockchain technology and visualization techniques. This methodology is technology agnostic, i.e., different blockchain technologies can be used for application implementation. The methodology can be summarized as follows:

(1)   Create the blockchain with the different network nodes, where each node corresponds to different users who will participate in data sharing. In our case study the nodes correspond to patients who decide to share their files as well as the buyers of information from these files.

(2)   Manage the transactions generated by different nodes. Here, we will focus on authentication, file transfer, and visualization. These transactions are combined with other transactions to create a new block.

(3)   Configure and customize the information to be visualized after choosing a tool for the network visualization.

(4)   Connect or integrate the blockchain with the visualization tool.

(5)   Demonstrate the visualization of how nodes are interacting during a transaction.

Figure 1 shows at a high level how the transactions are managed within the blockchain network.

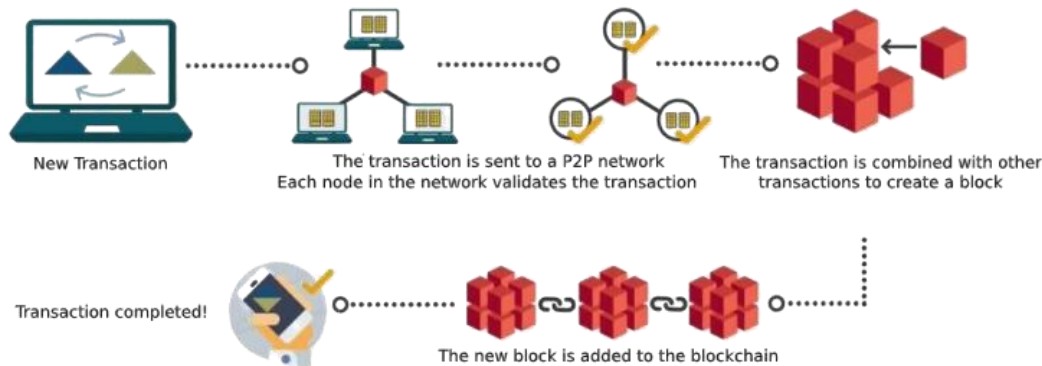

**Figure 1.** Methodology for transactions management within the blockchain network.

Figure 2 shows the basic blockchain structure. A blockchain is a data structure in which the information contained is grouped into sets of blocks. Each block has information on the previous block, and, using cryptographic techniques, this information can only be repudiated or edited by modifying all subsequent blocks. The information stored in each block includes: (a) records or transactions, (b) information about the block, and (c) a link to the previous block through a digital signature (hash). Each block has a specific and unmovable place within the chain, since each block contains information on the previous block as a hash. The entire chain is stored at each of the nodes that make up the network, so that all network participants have an exact copy of it. When a new record is created, it is verified and validated by all the nodes that form the network and then added to a new block and linked to the chain. Each node uses different types of certificates and digital signatures to verify information, as well as to validate transactions and data stored in the blockchain.

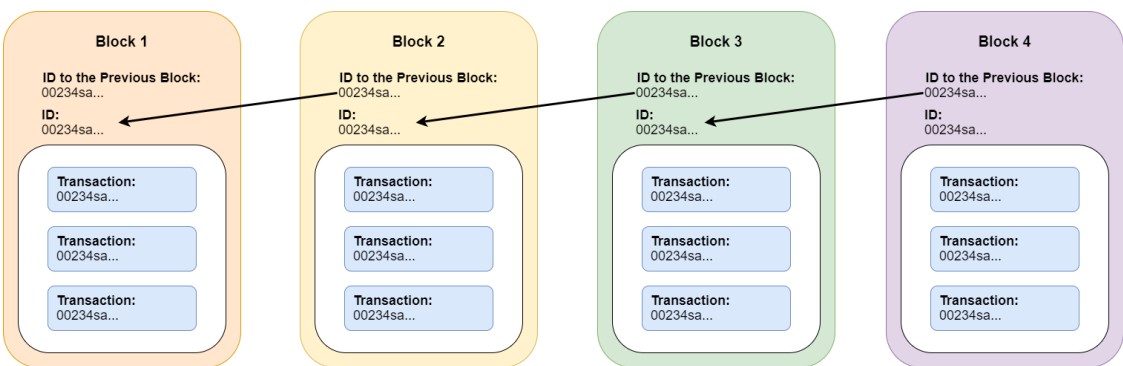

**Figure 2.** Basic structure of a blockchain.

Despite having introduced in this section the generic methodology based in blockchain, not all environments or organizations may use this methodology shown here. As Hebert et al. [64] point out, varying levels of security threats specific to a blockchain may call for an integrated multi-staged architecture. For the healthcare application chosen here, where a patient stores his or her data and provides access to these data to others, the methodology used here is considered appropriate. The data shared is tamper-proof because of the immutability property, and participants of the network are a-priori authenticated as trusted partners to share data using the network (i.e., permissioned blockchain). The application also uses the proposed transparency feature to enable users to see who accessed the data, and the blockchain keeps a record of every time data are accessed with a time stamp, thus providing an audit trail. The following section will discuss the application.

## 4. Case Study

The case study presented here allows patients to share their health data, including diagnoses and treatments, and gives research organizations access to these data for payment. The transparency of access and its purpose ensures that payments are made for the right purpose and accurately, while protecting patients' rights over their data. The payments increase patients' willingness to share their data for research purposes, and research institutions will benefit by paying a small amount to gather a large amount of patient data to support analysis. While such a payment of small amounts to patients may be viewed as too complicated [65], research organizations today spend large sums of money to solicit patient participation in clinical trials and a large proportion of this money goes to intermediaries. Within a blockchain, the patient has more control over their data and can monetize the data by selling it directly to potential buyers.

As the blockchain can track every access, the payment can be coupled with access, thus leading to immediacy and accuracy. Such transparency can lead to increased patient participation and improve the quality of clinical trials. Some systems have used token mechanisms for payments and several blockchains have their own crypto tokens. However, the tradability of tokens with fiat currency, liquidity, and the handling of inflationary pressures makes their use complicated [66]. Therefore, the system described here uses fiat currency (US dollars), thus creating a determined value for each transaction.

*Processes for Uploading and Accessing Health Data*

The main roles in the blockchain-based system are patients, caregivers, and buyers. The caregivers monitor patients for care continuity (e.g., doctors, external care providers, family members, etc.). The case study here, however, will focus on the interaction between patients and buyers, where patients upload health data and allow buyers to download and read the data after purchasing it. The patient data can be in the form of a Continuity of Care Document (CCD) or Fast Healthcare Interoperability Resources (FHIR). Patients get these data either from hospitals and clinics or can upload it from their

own devices (e.g., via Fitbit devices). These data can either relate to a patient visit to a care center (encounter) or an episode related to his health/wellness.

The data are uploaded one record at a time by the system front-end and is stored in the blockchain. The metadata about that data are stored in local storage, and this can include the nature of the data uploaded. Depending on the size of the data, this can take a couple of minutes. The patient is informed once the data are uploaded and stored. This is shown in Figure 3.

Patients publish the names of the files they wanted to share. When a buyer wants to purchase data, they are shown different types of data and the corresponding information (e.g., time range). Subsequently, once the buyer decides to purchase some data, the system determines the owner of the data and checks whether permission was provided. If permission was not already provided, the system informs the patient of the buyer request and the incentive offered by the buyer. If the patient provides permission, then the system stores the permission (for one patient, one buyer, and one piece of data) in the blockchain. It notifies the buyer of the permission, so the buyer can request the data to be read. The system stores the data access and deducts the payment from the buyer and credits it to the patient. The payments made are accumulated with each buyer read. These interactions are shown in Figure 4.

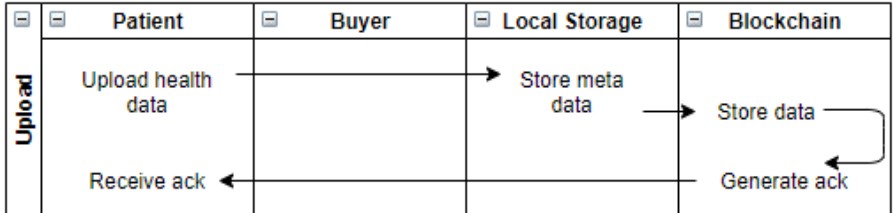

**Figure 3.** Process for uploading of health data by patient.

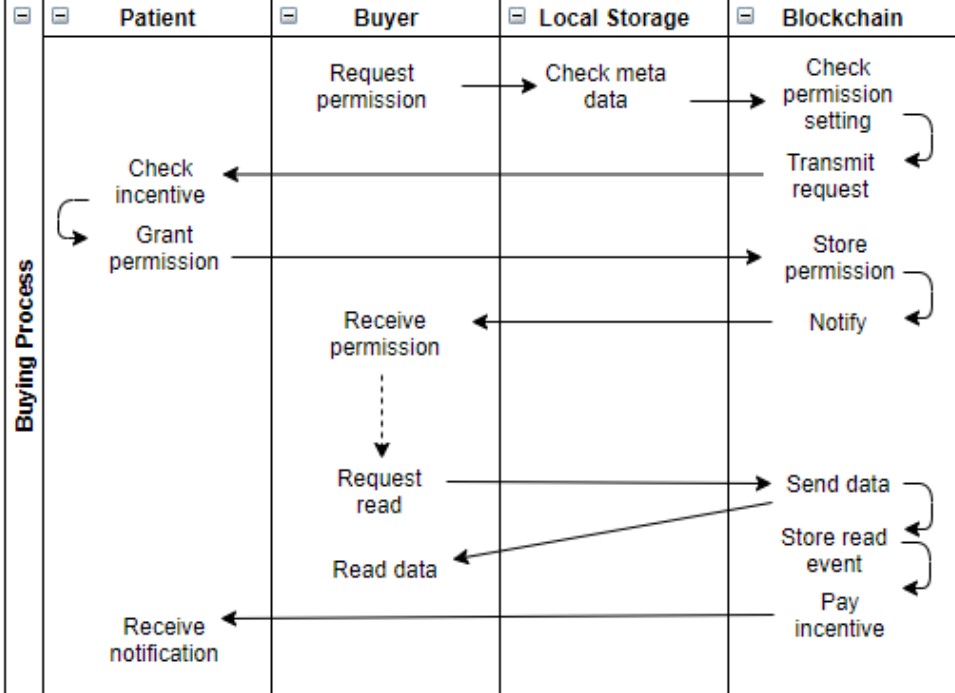

**Figure 4.** Process for access to health data by buyer.

The blockchain keeps health data, permission data, and the monetary amounts belonging to both the patient and the buyer. Based on the size of each block, these patient–buyer transactions can be spread across many blocks, as shown in Figure 5. The next section will discuss the implementation.

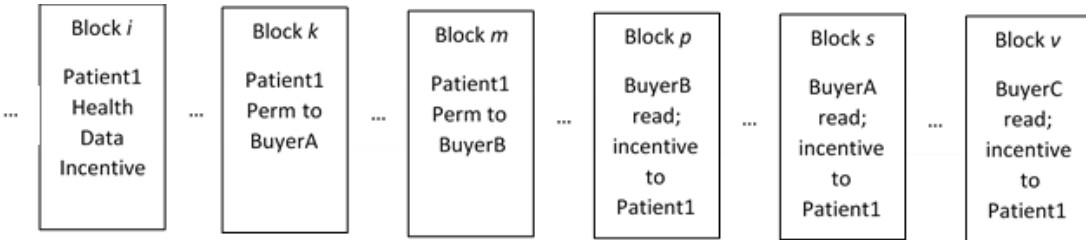

**Figure 5.** Patient's data and information across blocks.

## 5. Implementation

In this section, we will discuss the implementation of the case. It is assumed that permission is granted by the patient, his or her data are sent to the buyer, and these transactions are tracked.

The technologies considered for the development of this application were Corda R3, Hyperledger Fabric and Ethereum. After studying these three technologies, Hyperledger Fabric [67] was chosen for its robustness and the privacy it offers for the stored information compared to several competitors. It is also configurable, guarantees security, interoperability, and data sharing. Inside the Hyperledger family, there is also Hyperledger Sawtooth with a different consensus algorithm and a different mode of execution. For the purpose of this study, Hyperledger Fabric was chosen because its Explorer is much easier to use than the Explorer that comes with Hyperledger Sawtooth. The main challenge in the implementation was the integration of the different used technologies such as Hyperledger Fabric, Hyperledger Explorer or Vue.js; implementation details are shown in the following subsections.

### 5.1. Blockchain Creation

Each node in the network (associated with the users: patient, buyer, etc.) will be created using Vue.js. Different templates will be created for viewing files, sending files, and support authentication. When the permission is provided by the patient, the corresponding transaction and the subsequent block is created.

The first step calls for the downloading the blockchain platform using the latest version of Hyperledger Fabric from the official repository, unzip it and access the first-network folder to check accuracy of the download. Once in the folder and is running correctly, the message shown in Figure A1 will be displayed.

### 5.2. Transaction Management: Patient Permission, File Transmission and Block Creation

Once the permission is granted by the patient to share his or her data, the application will check that the recipient is in the system and the file is in the right format. Then, the file will be encrypted in base64. Base64 is a method of encoding and decoding binary data (e.g., HTML, CSS, text documents or images) [68]. After encrypting the file, the Application Programming Interface (API) endpoint will be called to upload the file to the blockchain. Subsequently, a "json" file with the user's credentials and the encrypted document is sent to the API. This information becomes part of the transaction and will be converted into a block, as shown in Figure 6.

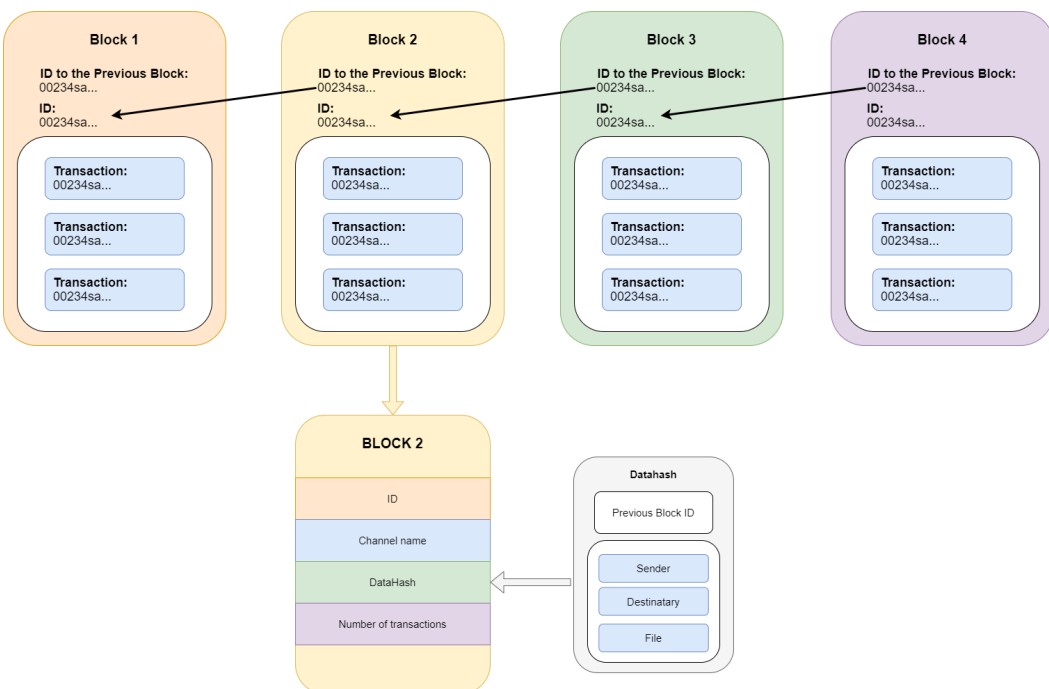

**Figure 6.** Description of information block.

## 5.3. File Reception

In this step, the receiver (the buyer) can download the shared files. When the receiver logs on to the home page and clicks "View my received documents", a screen (as shown in Figure 7) will appear. The recipient user will be able to download the documents needed, and these are ordered from the most recent to the oldest, showing the sender, the send date, and the ID of the sender. When the receiver clicks on "Download", the file will be decrypted using base64 and then downloaded.

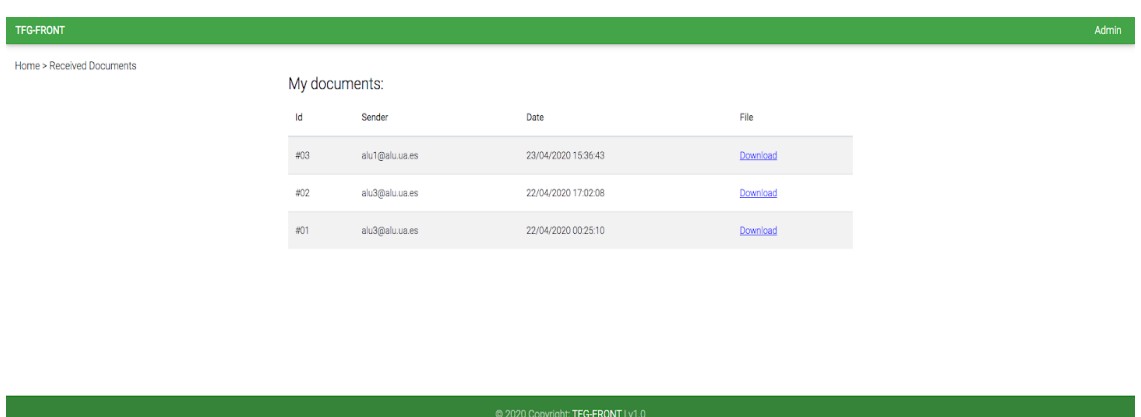

**Figure 7.** Received files.

## 5.4. Visualization Configuration and Connection with the Blockchain Network

Hyperledger Explorer will be used for the display of the network using React.js. It offers default templates ready to be launched or edited, and it provides several graphics to customize the templates for visualization. Such a method of sharing documents and using visualization to track its flow is useful in healthcare to build transparency and gain the trust of all actors involved. There are potentially other applications where such transparency is needed to ensure user adoption of blockchain technology

for sharing data. The rest of the section will discuss some of the implementation details such as the installation, configuration, and visualization of Hyperledger Explorer.

### 5.4.1. Installation and Configuration

For the Installation, the first step is to download the latest version of Hyperledger Explorer from the official repository, followed by downloading PostgreSQL packages, and running the database services to make sure the database has been installed correctly (as shown in Figure A2).

Once installed, the next step will be to authorize Hyperledger Explorer to access the network in Fabric (Configuration). In the "app" folder inside the main folder of "blockchain-explorer", the file "explorerconfig.json" should be modified (Figure A3).

In "platform", the fabric platform is used. In "PostgreSQL", the database credentials will be detailed. To connect Explorer with Fabric, access "blockchain-explorer/app/platform/fabric" where the file "config.json" will be modified. The goal here is to define the connection with Fabric (Figure A4). The name of the blockchain network in our case is set to "first-network".

Finally, we open the json file located at:

`/blockchain-explorer/app/platform/fabric/connection-profile/first-network.json`

Then, we update "adminPrivateKey", "signedCert" and "path" with the corresponding routes of the Fabric network for visualization (Figure A5).

Once Fabric and Explorer are connected, the last commands (Figure 8) are executed to build the project, which contains our case study:

```
./main.sh install (To make the build of the project.)

./main.sh clean (To clean up unnecessary files that were installed with the
     previous command.)

./main.sh test (To test the REST API as well as the interface components,
     it generates a document reporting errors.)
```

**Figure 8.** Commands needed to run Fabric.

### 5.4.2. Visualization

The final step is to visualize the blockchain network from an analytical point of view. For this purpose, it is necessary to modify some packages of Hyperledger Explorer. Its structure is shown in Figure A6.

In order to customize Hyperledger Explorer, the default code of the official package must be modified. It is developed with React.js and Redux frameworks. Therefore, to edit the components it is necessary to access the folder "/blockchain-explorer/client/src/components" and edit the components that are required. Here, we have only modified Charts, as it supports visualization. Figure 9 shows the dashboard of Explorer, including a set of panels with the current configuration.

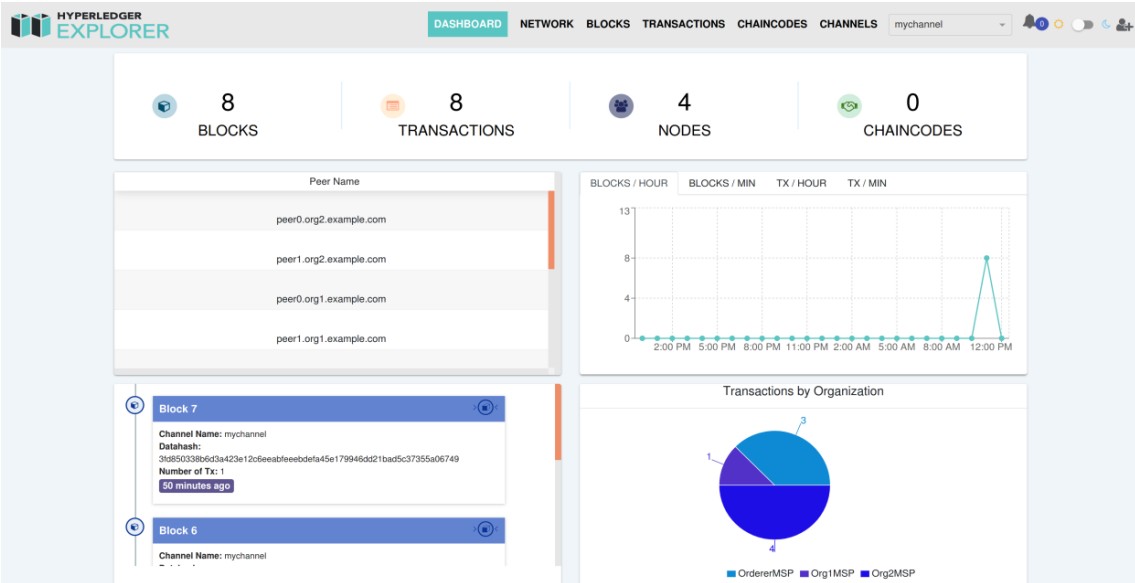

**Figure 9.** Dashboard of Hyperledger Explorer.

On the top panel, we can see that the network has eight blocks (from Block 0 to Block 7; the genesis block is a configuration block for a specific Hyperledger Fabric channel and contains no data) with eight transactions (one transaction per block). There are four nodes representing four different users registered on the network. In this case study, there are zero chaincodes since no smart contracts were created. Chaincode refers to the code for executing programs in the blockchain. These codes or smart contracts signify a particular mini agreement that gets automatically triggered when the condition values align to the required set of conditions. The word chaincode is a simple phrase to indicate that the code is related to the blockchain.

Below the top panel are the list of Peers on the left and the network traffic on the right. Peers are network elements that help maintain the network and verify and approve transactions. They also provide methods for interacting with the network, such as creating different APIs.

The component on the lower left shows the blockchain. It shows the last block added (Block 7). Each block has three different fields:

- Channel Name: The name of the channel through which the block has been created. A channel is a mechanism by which a set of components of a blockchain network interact and exchange information. They provide privacy to the network. There can be different channels, and users can access one or another, depending on how their permissions are configured.
- Datahash: This is an encrypted code that contains all the information of the block. Here, you can find information about the sender of the file, the receiver of the file, and the file itself.
- Number of Tx: This represents the number of transactions per block.

To the right of this last component is Transactions by Organization, an entity that has access to different channels and shows how network participants are grouped according to their privileges.

Finally, it is important to question the suitability of approaches similar to ours for inherently decentralized architectures such as distributed ledgers or blockchains, where processing, storage, and control flow are shared among many equal participants. Van Landuyt et al. [69] performed an analysis of blockchain security and the privacy of data it supports with other threat-modeling approaches discussed in the literature and their findings identify areas for future improvements needed for threat-modeling approaches.

### 5.5. User Study

A user study was carried out to determine the features important to users with respect to the visualization model and implementation. The total number of users within the authors' research group performing the study were 11: two full professors, three associate professors, three PhD students, and three degree students. Figure 10 shows the results from the users' responses. The users indicated that security and a user-friendly nature are the most important features. The preliminary results show that transparency in data sharing is important for user participation when there is no single trusted coordinating entity that users can rely on.

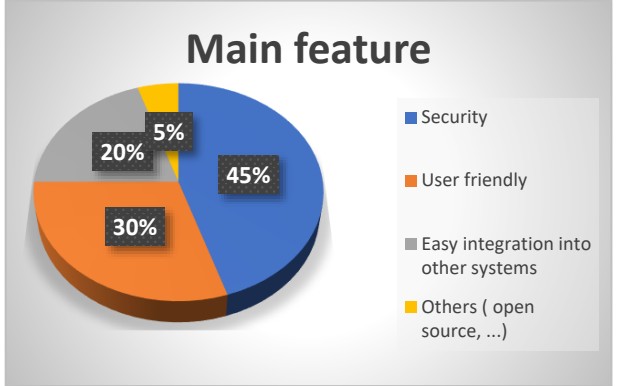
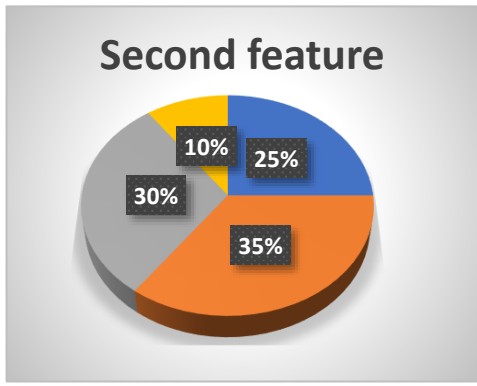

(**a**) Main feature to users with respect to the visualization model and implementation.

(**b**) Second feature to users with respect to the visualization model and implementation.

**Figure 10.** User study results.

In summary, the proposed digital platform can be used in any healthcare application where there are multiple actors (hospital, patients, external clinical and non-clinical care providers) sharing select data among each other to support care. The content of the file (or resource) to be shared and who it should be sent to is determined by the client (patient, provider, etc.), and the blockchain architecture supports interoperability among a number of distributed systems outside a hospital's own EMR. With the immutability of data stored and the authenticity of those accessing the data, the architecture ensures that those who are designated to receive the data are indeed the ones who are accessing the data. More importantly, by visually tracking the movement of data files, the users can see and interpret the activity. This is a key contribution of this paper.

## 6. Discussion

The implementation can be generalized to share different types of files based on the application context. For example, users may cast their vote on an issue or in an election and see how these are pooled by an authentic node on the network for compilation. Similarly, in today's COVID-19 environment, data from various test facilities and hospitals can be tracked for the number of people infected (or testing positive) and the number of hospitalizations and deaths for public health officials to develop regional patterns. With some of the demographic or geographical data of each node stored outside the blockchain, it can reduce the data redundancy but provide access to interpret the data traffic within the network. Moreover, blockchains using smart contracts can provide alerts in appropriate nodes based on data analysis. For example, an alert can be sent to a public health node on the network when the number of positive cases coming from that region exceeds a threshold for its regional population, so that it can develop alternative preventive practices. Similarly, it can trigger an alert to an emergency management vehicle station node when a hospital within its area has exceeded its hospitalization capacity, so that patients can be diverted to another hospital. Furthermore, some of these partners, such as emergency management vehicle stations or public health agencies, can be

outside the blockchain if they are primarily receiving alerts or aggregated data to reduce network complexity, or else in a separate blockchain that is used for receiving such alerts.

### 6.1. Future Research Directions

When care is moved outside a hospital and with a number of actors sharing different types of data at varying frequencies, future research needs to explore certain heuristics or algorithmic models to segment the digital platform that may include a mix of centralized and distributed networks. Each of these networks is synchronized to ensure that data moving within and across these networks are not lost. The larger the network, the greater the technical challenge of managing the actors and the data they share and the more complex the social challenge of aligning the goals of these actors. In addition, the distributed actors using blockchain must eventually interact with other actors (e.g., hospitals) who operate centrally coordinated patient health records or a government agency that regulates the type of data shared. This leads to three different possible research directions.

Addressing the technical challenge: Are there ways to decide when to segment the data based on frequency of use and the size of the data shared? Given the redundancy embedded in the way in which the blockchain replicates the data, decision rules may guide the size of the data to be shared, the type of data shared (e.g., images vis-à-vis text), the frequency of data sharing (e.g., once a month with a few nodes or real time for tracking infections) and, of course, the number of nodes who need access to these data. This may lead to the creation of subnetworks, which also are relevant in addressing the complexity of the social challenge.

Addressing the social challenge: Healthcare outside a hospital is supported by many different actors, such as clinical actors like pharmacies or testing labs, non-clinical actors like social workers or care givers of patients at home, or researchers who analyze data for treatment adherence or disease patterns. The motivation of these users to use such a platform to share data and the transparency they need to enhance their trust in using the system may vary. Therefore, having different networks support clinical actors, non-clinical actors, and analysis may lead to reducing the goal alignment complexity and help mitigate the need for visualization and associated complexities in system design. Moreover, many of these subgroups have varying levels of interaction with hospitals, thus creating the need for different gateways for data sharing with the hospital EMR, an issue which is discussed next.

Gateways to centralized systems: Hospitals and government agencies still drive much of healthcare around the world, and the type of data integration they need with external actors varies. For example, central public health agencies of regions or countries need aggregated data from hospitals and other external care providers like test facilities to track disease conditions, except during health emergencies when real time data access is critical. Similarly, hospitals may need certain data in real time from clinical actors outside the system like pharmacies to control over-prescription or use of drugs, whereas they need periodic data from social workers on patient adherence to treatment protocols. This means that each blockchain network may have to decide which centralized systems will become nodes and how data are aggregated and sent to these nodes based on pre-defined criteria. In some cases, the centralized systems may be part of a separate network, with the blockchain network of the distributed actors simply connected to the centralized system network to ensure the integrity of each.

### 6.2. Limitations

We reviewed the state of the art of both the challenges and opportunities offered by the blockchain technology-based solution in terms of modeling problems in general and in healthcare in particular. It is important to emphasize that although the technology itself is not new, the fundamental contribution of the paper here is the use of visualization to make blockchain use transparent. This highly model-driven and flexible methodology provides an integration with existing technologies, highlights various challenges and opportunities when they are integrated with a blockchain with IoT [70] and suggests improvements to support decentralization and scalability, identity (identification of every device), autonomy, reliability (verifying the data authenticity), security (validation by smart contracts among

other services), market of services (interesting solutions for an IoT ecosystem of services and data marketplaces), and secure code deployment (significant advantage of blockchain secure-immutable storage). Similarly, the survey [71] reviews blockchain challenges and opportunities and indicates a wide spectrum of blockchain applications extending from cryptocurrency, financial services, risk management and internet of things (IoT) to public and social services. The authors conducted a comprehensive survey on the blockchain technology with a focus on taxonomy, algorithms, applications, and technical challenges as well as recent advances to address some of these challenges.

Another important issue within the blockchain framework is cryptocurrencies, as they are an emerging economic force, but there are concerns about their security. The reason for this is due to the complex collusion cases and new threat vectors that could be missed by conventional security assessment strategies. Almashaqbeh et al. [72] propose an ABC: an Asset-Based Cryptocurrency-focused threat modeling framework, which demonstrates the effectiveness of some real-world use cases.

Finally, as we have observed in Section 5.5, the user study that has been carried out has the usual limitations of a preliminary study. For this reason, it will be necessary to extend it to a study with more users with different profiles in order to evaluate our proposal in a more exhaustive and comprehensive way and thus make it more general purpose. Finally, it would be necessary to compare our proposal with similar cases, where visualization is not present, to demonstrate the advantages of our methodology in gaining user trust to use a blockchain to share data.

## 7. Conclusions

This paper illustrates the use of a digital platform based on an underlying blockchain technology architecture to support data sharing by patients with external partners. It brings to surface the mechanism used by blockchain technology to send and receive data in a secure manner to engender trust among those sharing the data. Such transparency is key if the digital platform is to motivate patients, who are unfamiliar with the technology, to share their data with others who are willing to provide a service. Ultimately, the ease of use supported by interoperability among different patient and partner systems and the transparency with regard to how the data are shared among patients and partners are both critical for enhancing the external resources used to sustain care outside a hospital.

**Author Contributions:** Conceptualization, J.P., D.G., M.T. and P.K.; methodology, J.P., E.G., D.G., M.T. and P.K.; software, E.G.; validation, E.G. and P.K.; formal analysis, J.P., D.G., M.T. and P.K.; investigation, J.P., E.G., D.G., M.T. and P.K.; resources, E.G.; data curation, E.G.; writing—original draft preparation, J.P., E.G., D.G., M.T. and P.K.; writing—review and editing, J.P., D.G. and M.T.; visualization, E.G.; supervision, J.P., D.G., M.T. and P.K.; project administration, J.P., D.G. and M.T.; funding acquisition, J.P. and D.G. All authors have read and agreed to the published version of the manuscript.

**Funding:** This study has been partially funded by the ECLIPSE-UA project (RTI2018-094283-B-C32).

**Conflicts of Interest:** The authors declare no conflict of interest.

## Appendix A  Screenshots of Terminal Windows

**Figure A1.** Successful blockchain creation.

**Figure A2.** Check for correct database creation.

**Figure A3.** Hyperledger Explorer access to the network in Fabric.

**Figure A4.** Connection with Fabric: network name.

**Figure A5.** Connection with Fabric: path settings.

**Figure A6.** Hyperledger Explorer structure.

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
