# Peer review of "Using Visualization to Build Transparency in a Healthcare Blockchain Application"

_sustainability, doi:10.3390/su12176768_

Round 1

Reviewer 1 Report

The paper presents an approach of building transparency in healthcare blockchain applications through visualizations. The work seems interesting. However, I would like the authors to improve their work based on the following feedback.

The work seems alright in terms of modeling the problem. However, the scientific merit of the work is very poor. The authors did not clearly mention the challenges of the work. Also, the authors need to clarify why the existing works are not suitable for the problem.

The authors should target threat modeling based on the following articles:

Hebert, Cédric, and Francesco Di Cerbo. "Secure blockchain in the enterprise: A methodology." Pervasive and Mobile Computing 59 (2019): 101038.

Almashaqbeh, Ghada, Allison Bishop, and Justin Cappos. "ABC: a cryptocurrency-focused threat modeling framework." In IEEE INFOCOM 2019-IEEE Conference on Computer Communications Workshops (INFOCOM WKSHPS), pp. 859-864. IEEE, 2019.

Van Landuyt, Dimitri, Laurens Sion, Emiel Vandeloo, and Wouter Joosen. "On the Applicability of Security and Privacy Threat Modeling for Blockchain Applications." In Computer Security, pp. 195-203. Springer, Cham, 2019.

In the experiments, the implementation challenges should be clearly described and how the authors overcome those issues should be mentioned. What if we implement the model in another ledger technology? Are there any experimental results of it?

The authors used Hyperledger Explorer Structure to provide the exploration/visualization experience. This seems very much straightforward and there is no additional contribution except using an already existing tool.

The authors could have section for visualization section in the paper. They could explain the key aspects/challenges of visualization of this work. Later, they could implement the model by modifying the Hyperledger Explorer Structure.

The authors should also conduct an user study why and how this could be user friendly and conduct user acceptance test of the visualization model and implementation.

Other comments:

In Figure 2, there should be no red underlining Recv Notificn. The authors can produce better professional diagrams in “draw.io” online based software. Similar feedback for other figures.

Figures 4 and 5 can be squeezed/tightened to give it a professional look.

I did not find any sue of the appendix in the paper. Why it should be added in the paper is not clear.

Reviewer 2 Report

The paper proposes a new architecture to support the issues around sharing patient data (namely security and interoperability) in healthcare contexts that stagger the common hospital environment. The goal is to promote trust and transparency amongst major stakeholders.

The work has merit and is interesting, but seems to lack some structure and focus. 

  1. The title of the paper refers visualisation as a key way to achieve transparency but little to nothing is referred regarding implemented specificities of the case study (simply section 4.4). Despite there is a section that refers Hyperledger Explorer this can be further enhanced.
  2. This leads to one of my biggest concerns related to the content of the paper: how is visualisation the key in transparency? What is the data that allows authors to conclude this?
  3. I would advise the authors to first focus on the content and then revise and restructure the paper accordingly. I believe it would provide an adequate initial answer to the proposed research questions to show the case study architecture implementation depicted. This would imply that if the authors' decision is related to this, then the issue of visualisation must be addressed in future research. The authors can still present their work with Hyp Explorer, but they should consider removing the relevance of the topic from the title.

Major comments:

  1. Section 3 does not present the Methodology. The authors should choose to present a methodology and introduce it (for instance, the case study methodology). They should describe how they designed the study, for how long, where and how was the case chosen.
  2. Section 3 could also present the Case Study as it is currently doing, but in this case authors must change the section's title since the content is mainly related to a specific context and problem and the proposed solution for it.
  3. Section 4.1 should be named Architecture and should have a corresponding diagram or diagrams. The chosen technological platform (Hyp Fabric) should be somewhat introduced and explained. There are other options even inside the Hyperledger greenhouse such as Sawtooth. Were options considered? Why or why not? For instance, later in the document the authors refer 0 chaincodes because there are no smart contracts implemented. Not every reader has the knowledge of the wording/name used for smart contracts in Hyperledger Fabric. Plus, there are further details that should be explained (namely the channels and what Tx means).
  4. In the conclusions authors refer visualisation as a key contribution of the paper. If they choose to focus on this, then a bigger level of detail should be given to the visualisation aspects of the work.
  5. Lastly, despite being a part of the Introduction, I feel that the sustainability-agility reasoning given could be summarised and directed specific to the healthcare context.

Minor comments

  1. security >> when addressing this feature authors should consider clearly stating that blockchain is not used to enforce security nor does it address the issue.
  2. interoperability >> when addressing this feature authors should take care to avoid showing the "yet another system that needs to be seeded and has also interoperability issues" posture when describing their proposed solution
  3. data sharing >> when authors state "blockchain 181  technology allows for sharing multiple data types without forcing a single data normalization 182  method." they are referring to interoperability and not data sharing. Consider revising the bullets / options and joining some of them.
  4. IoMT >> describe the acronym and introduce its significance 
  5. In Figure 3 the last block has a "PRO incentive" that I was unable to understand - how is this different from the previous blocks?
  6. Include the reference "(Hyperledger: https://hyperledger-fabric.readthedocs.io/en/release-2.0/ accessed May 15, 2020)." as a normal reference instead of mixing it with the text
  7. Appendix B. does not show further information >> include the original article of Forbes 50 Blockchain 
  8. Figure 5 has too much blank space >> crop the image
  9. Figure 7. >> the description is "the network has seven blocks with eight transactions" >> does not comply with the image presented and later on the authors state "which is the eighth block since the first one is Block 0" >> so, what is it? 7 or 8?
  10. Appendix A has several failures such as "previous ones," >> previous one; "link to the previous and next block through" >> do the authors actually mean this?
  11. Consider a serious revision and proof reading of the paper before submitting it again. Typos and gramatical faults include, but are not limited to:
    • "widely 152  used in ensure integrity, confidentiality and accessible of medication information"
    • "allow buyers" >> allows  

Reviewer 3 Report

General comment

The paper is very interested and in general the different sections are well structured. The goal of the paper is to create a gateway between hospital systems and a network that connects patients and partners, in order to facilitate the sharing of patient data with different partners outside of a hospital network. The authors in this regard propose the use of block chain architecture to support the interoperability of data sharing between different systems.

Specific comments for sections

  • In Section 1. Introduction, the authors outline the research framework, namely the extension of the network of the business value chain to customers for co-creation of value and to partners to meet value, in a specific field of application the health sector. This sector is changing to improve its agility to meet the needs of patients and the sustainability of its resources, more than ever in the COVID-19 era. Interoperability of data sharing between different systems can efficiently respond to this issue and block chain technology can support peer-to-peer connectivity and meet the needs outlined. In this section the authors clearly set the objectives of this study and highlight how they will structure their paper.
  • In Section 2 the authors provide preliminary information on the use of block chain in multiple domains in the search field.
  • In Section 3 the authors present the general features of their block chain-based solution proposal.
  • In Section 4, authors apply their block chain-based solution to a case study, choosing to use Hyperledger Fabric to create the block chain network, Base64 for encoding and decoding binary data (for the Patient permission), the file "json" for the user credentials to be sent to the API together with the encrypted document files, according to the scheme shown in Figure 4. Then, the recipient user using Base64 can download all the information uploaded by the patient. In the end, authors choose to view the block chain network with Hyperledger Explorer.
  • In Section 5, the authors propose a brief summary of the results they have achieved in line with the objectives set.
  • In Section 6, the authors highlight the potential of this project to support the communication of information between patients and target users that may have a different nature. In addition, in this section, they highlight three different directions of development of this study, namely addressing the technical challenge, addressing the social challenge and gateways to centralized systems. These directions for the development of their work seem to me to be consistent with the objectives set.

Round 2

Reviewer 1 Report

The authors have addressed all the comments given in the previous round. I recommend to accept the current version of the paper.

Author Response

We appreciate the invaluable contribution of the reviewer, which has allowed us to produce a much higher quality paper.

Reviewer 2 Report

Thank you for the thorough review you have done on your work and for addressing all my comments. I consider the work has significantly improved.

Nevertheless, and despite understanding the reasoning that led to this output, I still have a couple of minor comments/suggestions.

1 - Consider moving all the Figures that are screenshots of terminal windows, to me included as appendix, namely Figures 6, 9, 10, 11, 12 and 13.

2 - Join sections 5.4.1 and 5.4.2 under a single section with the title "Installation and Configuration".

3 - Section 5.5. regarding the User study is lacking the information about the number of users that were submitted to the test. This type of study is also very limited. Thus this should be a part of the limitations of the work (section 6.2).

Finally, the version that you submit should be proofread once again. The version that was submitted had a couple of typos mainly due to the fact that it included the changes made in the document.
